# Modification of Whey Proteins by Sonication and Hydrolysis for the Emulsification and Spray Drying Encapsulation of Grape Seed Oil

**DOI:** 10.3390/pharmaceutics14112434

**Published:** 2022-11-10

**Authors:** Khashayar Sarabandi, Fardin Tamjidi, Zahra Akbarbaglu, Katarzyna Samborska, Pouria Gharehbeglou, Mohammad Saeed Kharazmi, Seid Mahdi Jafari

**Affiliations:** 1Department of Food Science & Technology, School of Medicine, Zahedan University of Medical Sciences, Zahedan 43463-98167, Iran; 2Department of Food Science & Engineering, Faculty of Agriculture, University of Kurdistan, Sanandaj 66177-15175, Iran; 3Department of Food Science, College of Agriculture, University of Tabriz, Tabriz 5166616471, Iran; 4Department of Food Engineering and Process Management, Institute of Food Sciences, Warsaw University of Life Sciences—SGGW, 02-776 Warsaw, Poland; 5Department of Food Science and Technology, Faculty of Nutrition and Food Science, Tabriz University of Medical Sciences, Tabriz 51656-65811, Iran; 6Nutrition Research Center, Tabriz University of Medical Sciences, Tabriz 51656-65811, Iran; 7Faculty of Medicine, University of California, Riverside, CA 92679, USA; 8Department of Food Materials & Process Design Engineering, Gorgan University of Agricultural Sciences and Natural Resources, Gorgan 49138-15739, Iran; 9Nutrition and Bromatology Group, Department of Analytical Chemistry and Food Science, Faculty of Science, Universidade de Vigo, E-32004 Ourense, Spain; 10College of Food Science and Technology, Hebei Agricultural University, Baoding 071001, China

**Keywords:** partial hydrolysis, structural modification, techno-functional properties, spray drying, encapsulated powders, oxidative stability

## Abstract

In this study, whey protein concentrate (WPC) was sonicated or partially hydrolyzed by Alcalase, then examined as an emulsifier and carrier for the emulsification and spray drying of grape seed oil (GSO)-in-water emulsions. The modification treatments increased the free amino acid content and antioxidant activity (against DPPH and ABTS free radicals), as well as, the solubility, emulsifying, and foaming activities of WPC. The modified WPC-stabilized emulsions had smaller, more homogeneous droplets and a higher zeta potential as compared to intact WPC. The corresponding spray-dried powders also showed improved encapsulation efficiency, oxidative stability, reconstitution ability, flowability, solubility, and hygroscopicity. The morphology of particles obtained from the primary WPC (matrix type, irregular with surface pores) and modified WPC (reservoir type, wrinkled with surface indentations), as well as the oxidative stability of the GSO were influenced by the functional characteristics and antioxidant activity of the carriers. Changes in the secondary structures and amide regions of WPC, as well as the embedding of GSO in its matrix, were deduced from FTIR spectra after modifications. Partial enzymolysis had better results than ultrasonication; hence, the WPC hydrolysates are recommended as emulsifiers, carriers, and antioxidants for the delivery and protection of bioactive compounds.

## 1. Introduction

Emulsifiers are the most common stabilizers in emulsions and emulsion-based food formulations. Due to consumers’ preference for natural and safer products, there is a growing demand for the replacement of synthetic surfactants with natural food emulsifiers. Proteins are the most important natural emulsifiers for food applications, and so far, the effectiveness of different proteins mainly originating from plants (e.g., soybeans, beans, wheat, rice, nuts, and oilseeds) and animals (e.g., milk, egg, meat, and fish) have been evaluated as emulsifiers or carriers of oil droplets [1]. Due to their amphiphilicity, surface activity, electrical chargeability, and polymeric nature, protein molecules minimize the emulsions’ destabilization mechanisms (e.g., flocculation, gravitational separation, coalescence, and Ostwald ripening) after dissolving and adsorbing onto the droplet surfaces [2].

However, low solubility, aggregation, agglomeration, sedimentation, and denaturation of proteins under acidic conditions and/or under mechanical and thermal stress may cause loss of their functional properties and limit their use as an emulsifier and carrier in various processes [3,4] Therefore, non-thermal techniques, including high-pressure, pulsed-electric field, cold plasma, irradiation, and ultrasonication, have been employed to modify the structural and functional properties of proteins [5].The effect of ultrasonication has mainly been attributed to the creation of bubble cavitation, and both shear and mechanical forces by which they destroy the protein aggregates and change the protein conformation (Jain and Anal, 2016) [6]. It has been shown that ultrasonication improves the foaming, emulsifying and antioxidant properties of proteins [7].

Enzymatic hydrolysis (enzymolysis) of proteins is also an important method to produce peptides with improved functional characteristics and biological activities [8,9] The function of various proteases is through the breakdown of peptide bonds and the production of low molecular weight (MW) fractions, which may show higher solubility, surface activity, and biological activity than the primary proteins [10]. Improving the solubility and surface activity of proteins (after structural modification) leads to an increase in amphiphilic characteristics, greater flexibility, and an increased tendency to be placed at the water/oil interface. This phenomenon plays an important role in stabilizing emulsions [5]. In addition, the film-forming ability and surface activity of proteins/peptides lead to these compounds being used as carriers for the microencapsulation of various bioactive compounds (polyphenols, anthocyanins, carotenoids, vitamins, etc.) [11,12,13]. Some peptides may show noteworthy health-promoting characteristics in the human body as antioxidants, antibacterial, anticancer, antihypertensive, and antithrombotic agents. Alcalase^®^ is a commercial and microbial serine protease widely used to produce protein hydrolysates with improved function. It has been documented that pretreatment of proteins with ultrasound waves enhances subsequent enzymolysis and improves the functional properties of the hydrolysates [6]

The emulsions and oil droplets that are kinetically stabilized with emulsifiers are still highly exposed to physicochemical destabilizing processes and pro-oxidants [14]. Promotion of lipid oxidation by destructive environmental conditions and the production of free radicals can lead to a decrease in the shelf-life and nutritional quality of food products and numerous health risks to consumers [15]. Solidification of emulsions increases the physicochemical and oxidative stability of protein-coated oil droplets or encapsulated bioactives within them. Spray drying is the most common technique for this purpose. By using different carriers in the solidification process of spray drying, all types of emulsions and oil droplets can be converted into powders with proper techno-functional characteristics [16,17].

Whey, the major by-product of the dairy industry, contains a protein mixture mainly comprised of β-lactoglobulin, α-lactalbumin, and serum albumin and is a good source of natural emulsifiers [7]. To our knowledge, there is no comprehensive research on the effect of ultrasonication treatment and enzymolysis on the structure, functions, and biological activity of whey protein concentrate (WPC). Hence, this study was aimed to investigate the amino acid composition, antioxidant activity, and techno-functional properties of WPC modified with ultrasonication or partial enzymolysis, and to examine its effectiveness as an emulsifier or carrier in the stabilization and encapsulation of grape seed oil (GSO) droplets within emulsions and during spray drying-solidification. Moreover, some characteristics of the emulsions and the finished spray-dried powders were examined. 

## 2. Materials and Methods

WPC powder (80% protein), Alcalase^®^ 2.4L, and GSO were purchased from Arla Co. (Central Denmark Region, Denmark), Novo Nordisk (Bagsvaerd, Denmark), and bio-natural (Castiglione di Sicilia CT, Italy), respectively. 2,2’-azino-bis(3-ethylbenzothiazoline-6-sulfonic acid) diammonium salt (ABTS), Comasi brilliant blue (G250), and1,1-Diphenyl-2-picrylhydrazyl (DPPH) were purchased from Sigma–Aldrich (St. Louis, MO, USA); all other chemicals used were purchased from Merck (Darmstadt, Germany). 

### 2.1. Modification of WPC 

WPC was sonicated or partially-hydrolyzed as follows: WPC was sonicated according to the method of Alizadeh and Aliakbarlu [18], with minor modifications. A 100 mL portion of WPC solution (5% *w/v*) in a phosphate buffer (0.01 M, pH = 7.4) was sonicated using a tip-type sonicator (UP200S, Hielscher, Teltow, Germany) for 15, 30, 45, or 60 min with a pulse pattern of 1.0 s on/1.0 s off (200 W, 24 kHz) and then freeze-dried (Christ, Germany; −20 °C and 0.1 mbar) and stored in air-tight bags at −18 °C until use. For enzymolysis, a solution of WPC (5% *w/v*), was partially hydrolyzed by Alcalase at a 100:2 ratio and at a pH of 9 in a water-bath shaking at 50 °C, for 30–120 min. Subsequently, the reaction flask was kept in a hot water-bath (95 °C for 15 min) to inactivate the enzyme, and centrifuged at 7000× *g* for 10 min. Finally, the hydrolysates-containing supernatant was collected, freeze-dried and stored at −18 °C [19].

### 2.2. Degree of Hydrolysis (DH)

A solution of each WPC sample in distilled water was mixed with a TCA solution (0.44 M) at a 1:1 *v*/*v* ratio, and after cooling in a refrigerator (4 °C; 10 min), was centrifuged (7000× *g*, 10 min). Then, the concentration of soluble proteins in the supernatant was determined according to the Bradford technique [20] using a series of standard bovine serum albumin solutions. DH (%) was calculated as follows: DH (%) = (mg protein in supernatant/mg protein in initial WPC solution) × 100(1)

### 2.3. Amino Acid Composition

The WPC samples were hydrolyzed at 110 °C with 6 N HCl for 24 h. The solutions were then purified by precipitation with 10% cold TCA, centrifugation (10,000× *g*, 15 min), and filtration through a 0.45-μm membrane. The free amino acid contents were measured using an HPLC system equipped with a reversed-phase column (Nova-Pak C18, 4 μm, Waters, Milford, MA, USA). The tryptophan content of samples was determined after alkaline hydrolysis. The amino acid composition was expressed as mg/g dry sample [21].

### 2.4. Antioxidant Characterization

#### 2.4.1. DPPH Free Radical Scavenging Activity

A 1.5 mL aliquot of each WPC solution (40 mg mL^−1^) was mixed with 1.5 mL of 0.2 mM DPPH solution, and after resting in the dark for 30 min, centrifuged (4000× *g*, 10 min) and the supernatant absorbance (A) was recorded at 517 nm [9]. The DPPH inhibition rate was calculated using Equation (2).
DPPH inhibition (%) = [1 − (A_sample_/A_blank_)] × 100(2)

#### 2.4.2. ABTS Free Radical Scavenging Activity

A 7-mM ABTS stock solution containing 2.45 mM potassium persulfate was prepared in distilled water, and after resting in the dark for 16 h, was diluted with 0.2 M phosphate buffer (pH = 7.4) to give an absorbance of 0.70 at 734 nm. Then, 30 µL of each WPC solution (10 mg/mL) was added to 3 mL of a diluted ABTS solution; after vortexing (30 s) and resting in the dark (6 min), the absorbance (A) was read at 734 nm [8]. The ABTS inhibition was calculated by Equation (2).

### 2.5. Techno-Functional Properties

According to [6] Jain and Anal (2016), the solubility, emulsifying activity index (EAI), emulsion stability (ES), foaming capacity (FC), and foaming stability (FS) of WPC samples were examined at different pH values. The water holding capacity (WHC) and oil holding capacity (OHC) were also measured by mixing 0.5 g samples with 5 g distilled water or soybean oil under magnetic stirring at room temperature for 30 min and then centrifugation at 4000× *g* for 20 min. The increase in sample weight after discarding the supernatant was expressed as WHC and OHC [22].

### 2.6. Preparation of O/W Emulsions

From the above-mentioned experiments, it was found that a 30-min time period is sufficient for the modification of WPC by sonication or enzymolysis to improve its antioxidant and techno-functional properties. Therefore, two modified WPC samples prepared under these conditions (Ul-30/En-30) were only examined, as emulsifiers or carriers, for the stabilization and encapsulation of GSO droplets within emulsions and during spray drying solidification (Appendix A).To prepare GSO-in-water emulsions, firstly, 16 g of each WPC sample was dissolved in 95 mL of a phosphate buffer (pH = 7.4) containing 0.01% sodium azide (as an antimicrobial agent) under continuous magnetic stirring at 40 °C for 2 h. The solution was then left at room temperature for 12 h for complete hydration. Finally, 4 mL of GSO was added drop-wise over 5 min into it, while under homogenization (Ultra-Turrax T50, IKA-Werke, Germany) at 7000 rpm, with additional homogenization for 10 min at 20,000 rpm.

### 2.7. Characterization of Emulsions

The mean droplet size (as Z-average), polydispersity index (PDI), and zeta potential of the emulsions were measured by DLS technique using a Zetasizer (NanoSizer 3000; Malvern Instruments, Malvern, UK) at a 25 °C and 90° angle. Before analysis, the samples were diluted 100 fold with water. The mean size of emulsions was monitored during 28 days storage at room temperature.

### 2.8. Spray drying Encapsulation

The encapsulation process was performed based on the method of Elik et al. [23], with some modifications. GSO emulsions were solidified in a mini spray dryer (Büchi B-290, Switzerland) under the following process parameters: inlet-air temperature of 130 ± 1 °C, outlet-air temperature of 80 ± 2 °C, feed rate of 5 mL/min, drying air flow rate of 0.56 m^3^ h^−1^, nozzle diameter of 0.7 mm, and an air pressure of 5.6 bar. The spray-dried powders were collected and packed in airtight bags, then kept in a refrigerator until use.

### 2.9. Characterization of Spray-Dried Powders

The production yield, moisture content, water activity (a_w_), bulk and tapped densities, flowability indices (angle of repose, Hausner ratio, and Carr index), solubility, wettability, hygroscopicity, and mean particle size of the spray-dried powders were measured according to the methods of Sarabandi et al. [24]. The surface morphology of air-dried emulsion droplets and also spray-dried powders was assessed using a Hitachi PS-230 scanning electron microscope (SEM) under a 25 kV accelerating voltage after coating with a gold layer. The color parameters (L*, a*, b*) of freshly prepared and spray-dried emulsions stabilized by WPC were determined with Image J software (NIH, Bethesda, MD, USA) from photos taken through a Canon digital camera (Powershot A3400). The hue angle and chroma value were calculated as follows:Hue = tan^−1^ (b*/a*) (3)
Chroma = [(a*)^2^ + (b*)^2^]^0.5^(4)

The encapsulation efficiency (EE) of GSO in spray dried powders was measured according to the method of Noello et al. [25], with some modifications. Briefly, a 2-g portion of each powder was dispersed in 20 mL of hexane, shaken manually for 10 min at room temperature, and filtered through a Whatman filter paper (no. 1), followed by rinsing of the residue three times with 3 × 10 mL of hexane. The filtrate solvent, which contained non-encapsulated (free) oil, was then evaporated at room temperature, followed by oven drying at 60 °C until it reached a constant weight. EE (%) was determined by weight difference using the following equation:EE (%) = [(Total oil − Free oil)/Total oil] × 100(5)

We assumed that the total oil in the powder and in the emulsion was equal as GSO is not volatile.

The FTIR spectra of all samples, after mixing with potassium bromide (ratio: 1:100) and pressing to take on a disk shape, were recorded consecutively using a FTIR spectrophotometer (Shimadzu 8400, Tokyo, Japan) in transmission mode over the wave number range of 400–4000 cm^−1^.

### 2.10. Lipid Oxidation and Storage Stability of Encapsulated Powders

The emulsions were reconstituted from the powders, and the extraction of GSO was performed. Samples were dissolved in a 30 mL solution of chloroform-glacial acetic acid (1:1, *v*/*v*). Then, 1.0 mL of a saturated potassium iodine solution was added by shaking for 1.0 min. The solution was kept for 3 min in darkness, and 100 mL of deionized water was added. The titration was started. Then, NaS_2_O_3_ (0.01 mol/L) was selected to titrate the liberated I_2_, and a starch solution (1%) was used as an indicator until the blue disappeared. The peroxide value (PV) after 0- and 4-weeks of storage was expressed as meq-hydroperoxides per kg of oil. PV of encapsulated and free GSO was calculated using the following formula [26]:PV (meq/kg) = (S − B) × C × 0.1269 × 78.8 × 1000/W(6)
where B is the titration of the blank (mL), S is the titration of the sample (mL), C is the molarity of the sodium thiosulfate solution, and W is the weight of the oil (g).

### 2.11. Statistical Analysis

The experiments were conducted in triplicate, and the data were reported as mean ± SD and analyzed by one-way ANOVA using SPSS software ver. 19.0 (SPSS Inc., Chicago, IL, USA). The Duncan’s test was employed to assess statistical differences (*p* < 0.05) between selected treatments.

## 3. Results and Discussion

### 3.1. Degree of Hydrolysis 

The DH is a measure of peptide-bond breakage rate in a protein affecting its MW, biological activity, and functional properties [21]. Intact-WPC had a DH of 1.8%, and sonication for 15, 30, 45, and 60 min, increased it to 3.7, 6.3, 7.2, and 7.8%, repectively; while enzymolysis for 30, 90, and 120 min, increased the DH to 8.5, 18.3, and 29.4%, respectively (*p* < 0.05). Other researchers have also reported that, an increase in enzymolysis or sonication time increases the DH of black bean protein [27], WPC [28], and chicken egg shell membranes [6].

### 3.2. Amino Acid Composition 

Table 1 shows the amino acid composition and/or the free amino acid (FAA) content of intact and modified WPCs. The amounts of hydrophobic FAAs (alanine, valine, isoleucine, leucine, tyrosine, phenylalanine, tryptophan, and methionine) were ~5.4, 10, and 28.5%, and the amounts of antioxidant FAAs (tryptophan, methionine, histidine, tyrosine, and lysine) were 2.1, 4.8, and 11.3% for intact-WPC, Ul-30, and En-30, respectively. The increase of hydrophobic FAAs in modified WPC could be an important factor in its improved functional characteristics, which is discussed in detail below. Also, the higher antioxidant FAAs in modified WPCs are a reason for their increased activity in the inhibition of DPPH and ABTS free radicals.

### 3.3. Antioxidant Activity

The scavenging activity of WPC for DPPH and ABTS free radicals is shown in Figure 1a,b, which first reacts with lipophilic and then with both hydrophilic and lipophilic radicals. The DPPH and ABTS inhibition rates of intact-WPC were 13.7 and 31.2%, respectively, and sonication for up to 45 min increased them to 25.1 and 47.3%, respectively; a longer duration of sonication (60 min) had no significant effect. The effect of sonication in improving antioxidant activity can be attributed to changes in the structural characteristics and surface hydrophobicity of proteins and the release of some peptides and FAAs, as a result of bubble cavitation, shear forces and local thermal shocks [29].

Compared to sonication, partial enzymolysis had a greater effect ion increasing the antioxidant activity of WPC. The maximum DPPH and ABTS inhibition activities (45.7 and 68.1%) were obtained for the WPCs hydrolyzed with Alcalase for 120 and 90 min, respectively (*p* < 0.05), and a longer enzymolysis (120 min) had no significant effect on the inhibition activity of ABTS. The enzymolysis of proteins increases their reactivity and releases hydrophobic and antioxidant amino acids [30]. Some amino acids neutralize or scavenge free radicals and stabilize diamagnetic molecules by donating protons or electrons [31]. Higher antioxidant activities after enzymolysis have also been reported for the proteins of mung beans [32], walnuts [10], *Spirulina platensis* [3], black beans [14], WPC [28], date kernels [30], and grass turtles [33]. 

This study was continued only with the modified WPC treated with ultrasound or enzyme for 30 min (Ul-30, En-30), since it was aimed to compare the effectiveness of mild or minimal levels of these treatments in improving the properties of WPC as an emulsifier or carrier for GSO. Moreover, it has been reported that severe enzymolysis reduces the techno-functional characteristics of protein hydrolysates by decreasing their amphiphilicity, unfolding ability, surface activity, and reorientation at the interface [34].

### 3.4. Techno-Functional Properties

Solubility is a basic requirement for proteins to demonstrate other functional properties such as emulsifying, foaming, and gel-forming abilities [33]. Therefore, the solubility (Figure 2) and other functional characteristics (Figure 3) of WPCs were investigated at different pH values. The solubility, emulsifying, foaming ability, and foam and emulsion stability indices for Ul-30 and En-30 were significantly increased in intact-WPC, especially at acidic pH values near the isoelectric point of WPC (pI ~ 5). But, at neutral or basic pH values, no significant improvement and/or even a loss in solubility and functional characteristics was observed. The partially hydrolyzed WPC (En-30) had better functional characteristics than the sonicated sample (Ul-30). However, the longer hydrolysis times and the decrease in the molecular weight of peptides gradually caused the loss of functional properties (emulsification and emulsion stability, foaming and foam stability) (Appendix A). 

The increase in solubility of Ul-30 can be correlated to the decrease in protein particle size because of the breakage of weak non-covalent bonds and the dissociation of protein aggregates; the exposure of some internal sulfhydryl groups, as a result of cavitation shear forces and micro-streaming, also contributed to an increase in the solubility of U1-30 [5]. The higher solubility of WPC after enzymolysis can be related to the breakage of disulfide bonds, the dissociation of insoluble protein aggregates, a decrease in MW, an increase in the exposure of hydrophilic regions, and the feasibility of their interaction with water [35]. The decrease in protein bodies’ size and the improvement of their solubility, conformational characteristics, and surface activity are some reasons for the higher emulsifying and foaming capacities of WPC after sonication or enzymolysis [6,36]. However, other factors, such as the alteration in amphiphilicity and surface charge of modified proteins as a result of pH change, also affect these characteristics [33]. The loss in techno-functional properties of modified WPC at neutral and higher pH values (Figure 3) confirms this statement.

### 3.5. Physical Characteristics, Morphology and Stability of Emulsions

The intact and modified WPCs were used as an emulsifier for GSO in water emulsions (Figure 4a–c), for which mean droplet size, PDI, and zeta potential data are shown in Table 2. The En-30, and then Ul-30, samples were more effective than intact WPC in the emulsification process, since the emulsions they yielded were smaller in size, had a lower PDI, and had higher zeta potential values. The increase in solubility, molecular flexibility, surface activity, and emulsifying ability of modified WPCs makes them more appropriate for binding onto and positioning at the oil-water interface, reducing interfacial tension, and exposing some efficient groups and charged amino acids [7,10,37]. The SEM images of emulsions showed relatively spherical particles with smooth surfaces and a size in accordance with DLS data (Figure 4d–f). 

The mean droplet size of emulsions was monitored during the 4-week storage period, in the stability test (Figure 4g). Emulsions stabilized with En-30 showed a minimal increase in size and were found to be the most stable, while the one containing Ul-30 showed a maximal increase in size, although it had a smaller size than intact-WPC after production. The protein hydrolysates create high surface activity and zeta potential values, which are highly efficient in forming a stable film around the oil droplets and reducing the rate of destabilizing mechanisms such as gravitational separation, coalescence, phase separation, and creaming [2]. Although, the sonicated and intact proteins may not be able to form a stable film around the oil droplets. Figure 4h shows an SEM image of the aggregated oil droplets in the emulsions produced with Ul-30, after 4-weeks of storage. In a previous study [19], the emulsions stabilized with rice protein hydrolysates had a smaller size, a greater zeta potential (−52 mV), and a higher stability than those stabilized with natural protein (−44 mV).

### 3.6. Physicochemical Characteristics of Spray-Dried Emulsions

Table 3 shows the properties of powders obtained from the spray drying of GSO emulsions stabilized with different carriers (emulsifiers), including intact and modified WPCs. The powder production yield was >65%, indicating a successful process of emulsion solidification to powder by spray drying. The low levels of moisture content (3.6–3.2%), and a_w_ (0.27–0.23) of the powders indicate that they are resistant to microbial growth. The low a_w_ of powder from En-30 emulsions indicates the presence of a large number of soluble molecules that result from proteolysis. The solubility of powders produced with En-30, Ul-30, and intact-WPC decreased in this order, which is in accordance with the order of carriers’ solubility. The hygroscopicity of powders from En-30 was greater than that of Ul-30 or intact WPC. This can be due to structural changes in the enzyme-treated WPC, which lead to high accessibility, exposure of hydrophilic regions, and high solubility [19]; the decrease in the surface oil of the final powder can also be another reason for this result. Modification of WPC with ultrasound or enzyme increased the particle density of the spray powders. This can be attributed to differences in structural characteristics, surface morphology, size and size distribution, adhesion, agglomeration, and caking behaviors of particles which lead to different porosities [25,38]. The powders had an appropriate flowability without being affected by the carrier modification [8].

### 3.7. Color and Morphology of Powders

Table 4 shows the color parameters of freshly prepared emulsions and their corresponding spray-dried powders. Treatment of WPC with ultrasound or enzymes increased the lightness (L* value) of liquid emulsions, not their powders. Modification of WPC with ultrasound or enzyme led to a decrease in redness (a*), yellowness (b*), chroma values, and an increase in the Hue angle of liquid and spray-dried emulsions; the effect of modification treatments was not the same in this regard (Figure 5). It has been documented that both the carrier nature and the chemical reactions such as pigment degradation, lipid oxidation, and browning reactions influence the color parameters [39]. In another study, enzymolysis of squid by-products protein with Flavourzyme^®^ led to a decrease in the lightness, redness, and yellowness of the freeze-dried hydrolysates [40].

The SEM images showed that morphological characteristics were significantly affected by the modification of the carrier (Figure 5). The powder produced with intact WPC had irregularly wrinkled particles with smooth surfaces and many indentations; the presence of numerous pores on the particles’ surfaces and hollow structures (matrix-type) could also be recognized (Figure 5b). While, the powder produced with Ul-30 had shapeless, irregular, clustered, and agglomerated particles (Figure 5e), indicating that the process of microcapsule formation was poor. On the other hand, the powder of En-30 had relatively spherical particles with surface grooves and both internal and external small sporadic pores (reservoir-type) (Figure 5h). It has been reported that the chia oil emulsions stabilized by modified starch WPC had spherical particles with irregular surfaces and different sizes, while those stabilized by modified starch WPC pectin had slightly spherical particles with high roughness. Moreover, the particles obtained from the two emulsions showed no surface cracks or grooves but contained small pores, indicating dispersion of the core material (oil) as small droplets into the wall matrix [25]. Apart from the type and concentration of components, the surface characteristics and morphology of particles are also affected by the moisture evaporation rate, solids transfer to the wall layer, and crust formation during the spray drying process [41].

### 3.8. Encapsulation Efficiency, Reconstitution, and Oxidative Stability of Spray Dried Powders

EE is one of the most important factors affecting the oxidative stability of spray-dried oil powders during storage [42]. The EE of powders ranged from 83 to 91%, and modification of WPC with ultrasound and especially enzymes, led to a higher EE (Table 5). This result can be attributed to the exposure of hydrophobic groups buried in the interior of protein molecules after enzymolysis (or sonication), which leads to an increase in their affinity for the hydrophobic core material and their emulsifying activity, thereby resulting in a higher EE [7,37,43]. In another study, rice protein was hydrolyzed to different DHs (1 to 10%) by Flavourzyme; only the hydrolysate with a 10% DH led to an increase in the EE (by 6.8%) of spray-dried linseed oil [19].

The average size of spray-dried powders ranged between 8.6 and 9.8 μm, and pretreatment of the carrier (WPC) with ultrasound or enzyme decreased the particle size. The size of spray-dried powders is affected by factors such as carrier concentration, process temperature, and feed viscosity. Compared to intact proteins, the modified ones may reduce the size of spray-dried particles by reducing the feed viscosity [25,41]. Particles with larger sizes usually have a higher EE due to having a smaller surface area [39]; but in this study, although the modified WPCs yielded smaller particles, they also increased EE. The size of reconstituted emulsions from spray-dried powders was 54 to 157% greater than those of primary emulsions. Pretreatment of WPC with enzyme or ultrasound reduced particle growth during the spray drying-reconstitution cycle, and enzymolysis was more effective. The increase in reconstitution ability of powders is a result of improvements in the functional properties (e.g., solubility, emulsifying activity, and stable film-forming) of their protein carriers by modification treatments [36]. 

A key goal of encapsulating oils enriched in unsaturated fatty acids is to increase their oxidative stability against destructive environmental factors. The PV for GSO in all freshly spray-dried emulsions was <5 mEq peroxide/kg oil, and was minimal for emulsions stabilized with En-30 (*p* < 0.05). After 4-weeks of storage, PV in spray-dried powders increased to >60 mEq peroxide/kg (Figure 6a). The powders prepared with intact-WPC and En-30 showed the highest and lowest PV, respectively. The low EE, the presence of hollow structures, and numerous pores on microparticles and the high particle specific surface area created by indentations can limit the oxidative stability of GSO in intact WPC (Figure 6b). While the higher stability of GSO in En-30 can be attributed to the improved functional characteristics (e.g., barrier film formation) and antioxidant and metal-chelating activities of the hydrolysates [39], along with the relatively uniform dispersion of GSO droplets in their matrix and the formation of matrix-type microparticles (Figure 6c). It should be remembered that the decrease in structural stability and integrity of emulsions due to shear and thermal stresses imposed on emulsifiers during spray drying encapsulation is still an important reason for the high surface oil or low EE and limited oxidative stability of spray-dried emulsions [19].

### 3.9. FTIR Data

Figure 7 shows the FTIR spectra for intact and modified WPCs, as well as for their corresponding spray-dried emulsions. Some structural changes in modified WPCs were deduced from FTIR data as follows: (i) the peak intensity in the amide A region (N-H and O-H stretch bonds) changed and slightly shifted (from 3296 to 3299 cm^−1^); (ii) the peak in the amide B region (C-H and O-H stretch bonds) also showed a minor shift (from 2927 to 2928 cm^−1^). These changes can be correlated to protein unfolding and subsequent interaction via exposed hydrophobic side chains or oppositely-charged groups, and the formation of new H-bonds [44]; (iii) the peak intensity at 1650, 1536, and 1241 cm^−1^ (corresponding to the regions of amides I, II, and III, respectively) changed, and some peaks also appeared at 1650, 1537, and 1243 cm^−1^. Most of these changes are related to the amide II region and to the NH groups buried in hydrophobic regions of proteins. Moreover, a perturbation in the vibration of residues (phenylalanine and tyrosine) may be caused by the unfolding and protonation of proteins [45]; (iv) the peak intensity at 1074 cm^−1^ (for C=O stretch in polysaccharides) changed and shifted to 1073 and 1076 cm^−1^, reflecting the exposure of some sugar regions that had previously been buried in carriers [36]. 

In the FTIR spectra of spray-dried emulsions prepared with modified WPCs, the higher peak intensities of the amide A and amide B regions confirm the occurrence of hydrophobic interactions due to the embedding and dispersing of GSO in the microparticles. Also, the occurrence of a peak at 1743 cm^−1^ and the increase in its intensity can be attributed to the interaction of the oil with carboxyl groups in the protein. In another study [26], a shift in the peak for the core material (walnut oil), from 1725 to 1750 cm^−1^, was found after encapsulation in a soy protein isolate, which was attributed to embedding of the oil into the carrier structure. Moreover, it has been reported that a new stretching band appeared at 3285 cm^−1^ for pepper seed oil, after encapsulation in Arabic-maltodextrin gum, and the band for stretching vibration of cis double bonds at 2926 cm^−1^, as a molecular fingerprint for oil, does not appear in the carriers’ spectra [42]. Another important band, associated with the ester carbonyl functional group of triglycerides, appeared at 1745 cm^−1^ [46]. The appearance of main bands of core material (oil) in the spectra of powders indicates a successful encapsulation process without any noticeable chemical interactions between the core and wall compounds [42].

## 4. Conclusions

Currently, many researchers are targeting the selection of natural carriers for successful encapsulation and solidification of bioactives using the most traditional and economical technique, i.e., spray drying. Proteins can be used as both emulsifiers and carriers during emulsification and subsequent spray drying of bioactives. However, their functional properties are affected by feed characteristics (e.g., composition and acidity) and spray-drying conditions (e.g., thermal and shear stresses). In this study, the structural, antioxidant, and techno-functional properties of WPC were improved by sonication or partial enzymolysis for 30 min. The GSO-in-water emulsions stabilized with modified WPCs showed proper physical attributes, and after spray drying, they yielded powders with improved physicochemical and morphological properties, oxidative stability, and reconstitution ability. In this regard, the partially hydrolyzed WPC was more effective than sonicated samples. The structure and functional properties of natural biopolymers, such as WPC, can be improved with non-chemical green methods (e.g., sonication and enzymolysis) before utilization as an emulsifier, carrier (encapsulant), and texture modifier in the food, pharmaceutical, and cosmetic product industries.

## Figures and Tables

**Figure 1 pharmaceutics-14-02434-f001:**
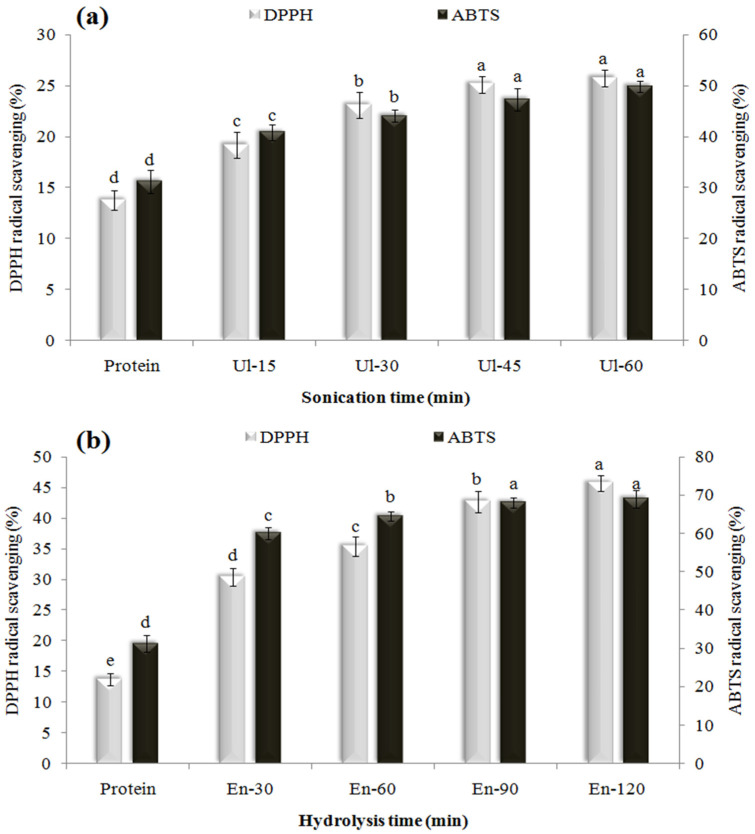
Effect of sonication time (**a**) or enzymolysis (**b**) on the DPPH and ABTS radical scavenging activity of WPC. Data are presented as mean (*n* = 3) with standard deviation (the same letters above bars indicate that the differences between average values were statistically not significant *p* < 0.05).

**Figure 2 pharmaceutics-14-02434-f002:**
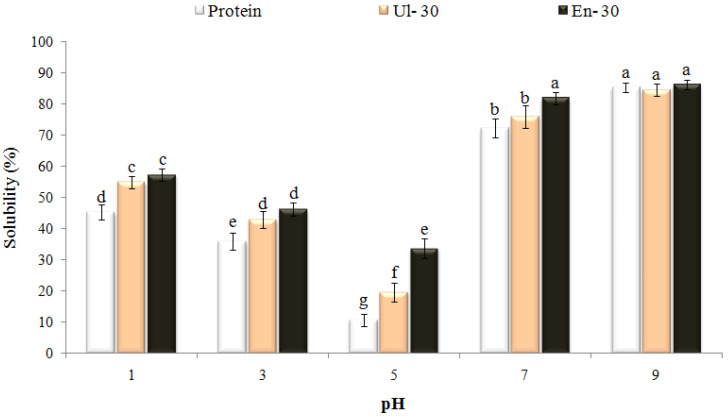
Effect of pH on the solubility of WPC modified by sonication or enzymolysis for 30 min (Ul-30/H-30). Data are presented as mean (*n* = 3) with standard deviation (the same letters above bars indicate that the differences between average values were statistically not significant *p* < 0.05).

**Figure 3 pharmaceutics-14-02434-f003:**
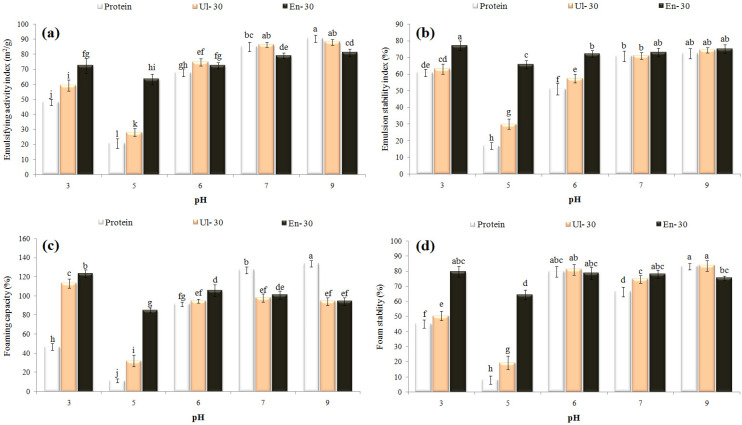
Emulsifying activity (**a**), emulsion stability (**b**), foaming capacity (**c**), and foaming stability (**d**) for intact WPC and WPC modified by sonication or enzymolysis for 30 min (Ul-30/H-30), at different pH values. Data are presented as mean (*n* = 3) with standard deviation (the same letters above bars indicate that the differences between average values were statistically not significant at *p* < 0.05).

**Figure 4 pharmaceutics-14-02434-f004:**
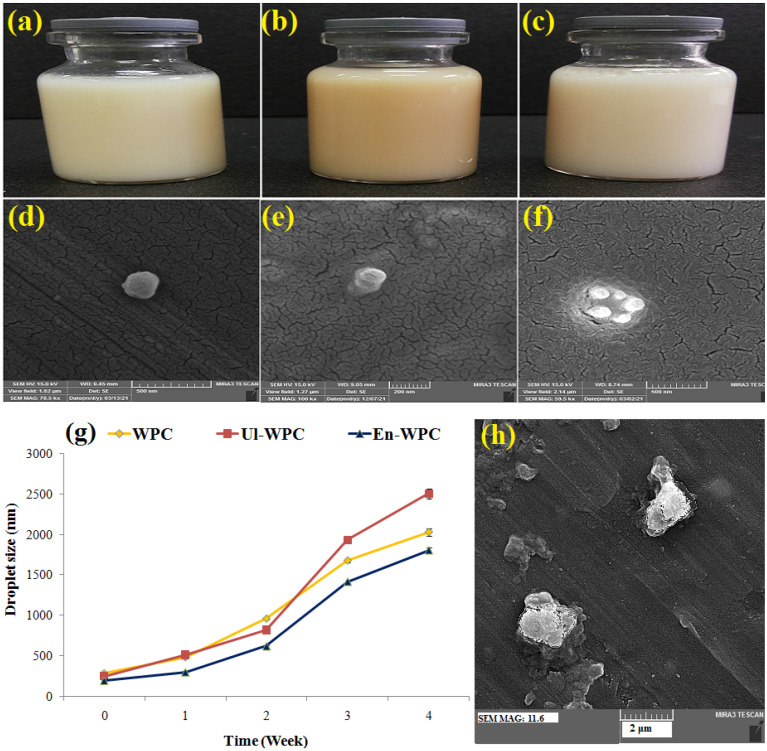
Optical (**a**–**c**) and SEM (**d**–**f**,**h**) images, and droplet size vs. storage time plot (**g**) related to grape seed oil emulsions stabilized with Ul30-WPC (**a**,**d**,**h**), intact-WPC ((**b**,**e**), and En30-WPC (**c**,**f**).

**Figure 5 pharmaceutics-14-02434-f005:**
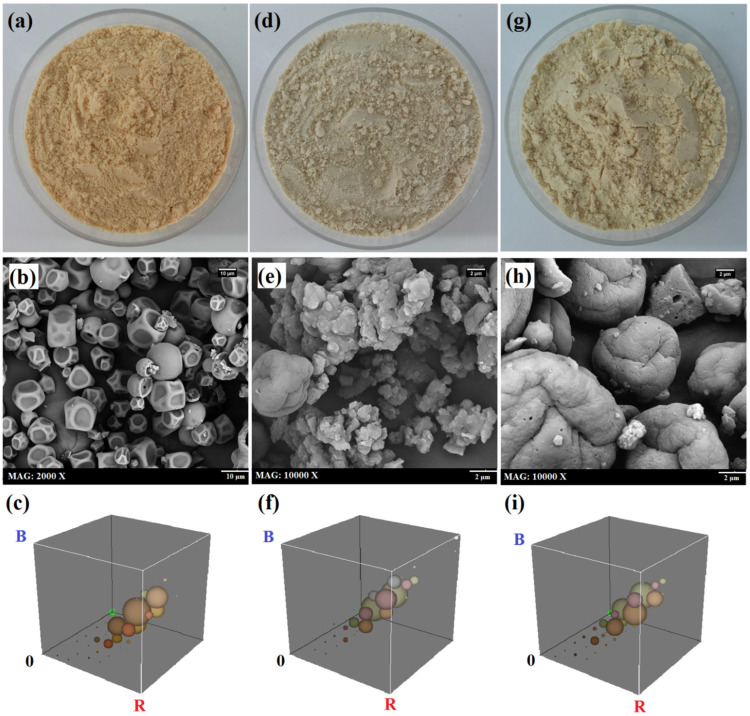
Optical and SEM images, and RGB plots related to spray-dried grape seed oil emulsions prepared with intact WPC (**a**–**c**), WPC modified by sonication (**d**–**f**), or enzymolysis (**g**–**i**) for 30 min.

**Figure 6 pharmaceutics-14-02434-f006:**
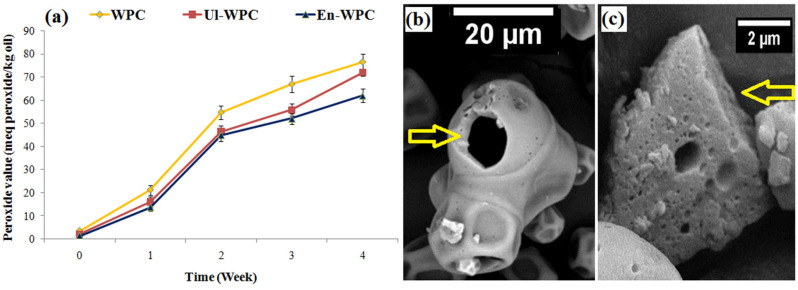
Peroxide value vs. storage time plot (**a**) and SEM images (**b**,**c**) of spray-dried grape seed oil emulsions prepared with intact WPC and WPC modified by enzymolysis for 30 min (En-30).

**Figure 7 pharmaceutics-14-02434-f007:**
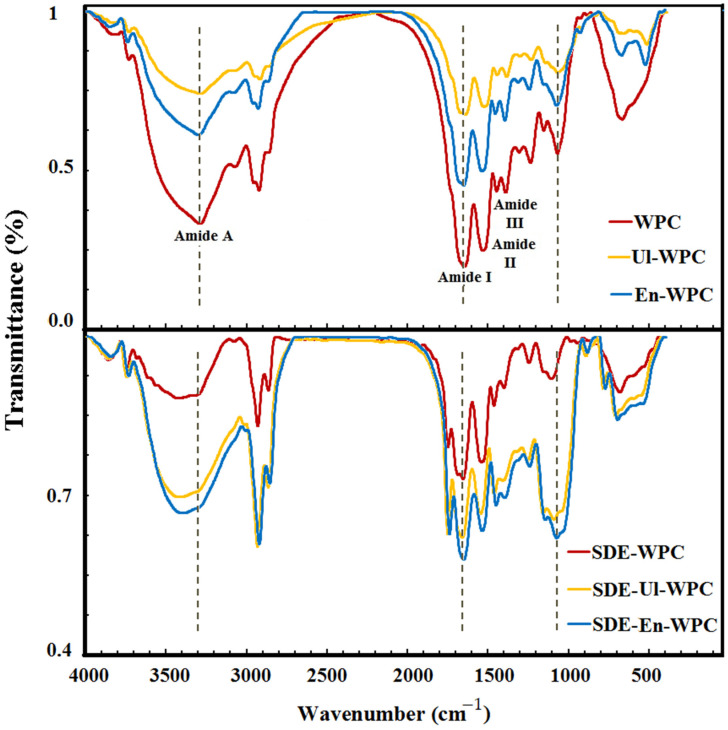
FTIR spectra for (**top**) intact WPC and WPC modified by sonication or enzymolysis for 30 min (Ul-30/En-30), and (**bottom**) spray-dried grape seed oil emulsions.

**Table 1 pharmaceutics-14-02434-t001:** Changes in the free amino acid (FAA) content of WPC during enzymolysis (En) and ultrasonication (Ul).

Amino Acid	Crude Protein (TAA)	FAA (mg Amino Acid/g Dry Sample)		
		0 min	30 min-Ul	30 min-En
Aspartic acid	77.1	0.5	2.5	4.1
Glutamic acid	128.9	2.9	3.9	5.8
Histidine	6.4	-	0.8	2.1
Serine	44.5	1.1	1.7	3.6
Arginine	22.6	0.3	2.6	4.5
Glycine	45.4	0.8	2.3	3.9
Threonine *	56.3	0.1	2.1	4.3
Alanine	44.7	0.2	1.9	3.8
Tyrosine	15.3	0.1	1.1	1.5
Methionine *	27.8	0.4	1.3	2.4
Valine *	15.9	0.1	0.8	2.7
Phenylalanine	29.7	0.3	1.2	3.6
Isoleucine *	40.3	1.4	1.3	5.1
Leucine *	87.2	2.8	2.4	8.3
Lysine *	71.3	1.5	1.3	4.2
Tryptophan *	14.2	0.1	0.3	1.1
HAA	275.1	5.4	10.0	28.5
AAA	135.0	2.1	4.8	11.3
TAA	727.6	12.6	27.5	61.0

* denotes essential amino acids; Hydrophobic amino acids (HAA) = alanine, valine, isoleucine, leucine, tyrosine, phenylalanine, tryptophan, and methionine; Antioxidant amino acids (AAA) = tryptophan, methionine, histidine, tyrosine, and lysine; Total amino acids (TAA).

**Table 2 pharmaceutics-14-02434-t002:** Mean droplet size, PDI, and zeta potential of emulsions stabilized with WPC and its modified forms.

Treatment	Size (μm)	PDI	Zeta Potential (mV)
WPC	0.286 ± 9.61 ^a^	0.33 ± 0.02 ^a^	−8.56 ± 1.15 ^a^
Ul-30	0.253 ± 9.07 ^b^	0.31 ± 0.02 ^ab^	−10.70 ± 1.34 ^a^
En-30	0.192 ± 10.02 ^c^	0.28 ± 0.02 ^b^	−14.03 ± 1.53 ^b^

Data are presented as mean ± standard deviation (*n* = 3), and different letters in the same column indicate significant differences at the 5% level in Duncan’s test.

**Table 3 pharmaceutics-14-02434-t003:** Physicochemical and techno-functional properties of spray dried grape seed oil powders.

Sample	Yield (%)	Moisture (%)	a_w_	Solubility (%)	Hygroscopicity (%)
WPC	68.73 ± 2.01 ^a^	3.57 ± 0.32 ^a^	0.267 ± 0.01 ^a^	83.80 ± 1.81 ^c^	13.63 ± 0.91 ^b^
Ul-30	65.23 ± 2.74 ^a^	3.37 ± 0.15 ^a^	0.243 ± 0.01 ^ab^	87.63 ± 1.98 ^b^	15.17 ± 0.99 ^b^
En-30	65.23 ± 2.74 ^a^	3.20 ± 0.20 ^a^	0.227 ± 0.01 ^b^	92.70 ± 1.70 ^a^	18.47 ± 1.02 ^a^
	**Bulk Density (g/mL)**	**Tapped Density (g/mL)**	**Angle of Repose (°)**	**Hausner Ratio**	**Carr Index**
WPC	0.487 ± 0.01 ^b^	0.600 ± 0.01 ^c^	24.33 ± 1.53 ^b^	1.230 ± 0.01 ^a^	0.187 ± 0.01 ^b^
Ul-30	0.533 ± 0.01 ^a^	0.670 ± 0.01 ^a^	30.00 ± 2.00 ^a^	1.257 ± 0.01 ^a^	0.207 ± 0.01 ^a^
En-30	0.520 ± 0.01 ^a^	0.650 ± 0.01 ^b^	27.33 ± 1.53 ^ab^	1.247 ± 0.01 ^a^	0.203 ± 0.01 ^a^

Data are presented as mean ± standard deviation (*n* = 3), and values denoted by different letters within each column are significantly different (*p* < 0.05). WPC: Whey protein concentrate; Ul: Ultrasonicated; En: Hydrolyzed.

**Table 4 pharmaceutics-14-02434-t004:** Color properties of primary and spray dried grape seed oil in water emulsions.

	L*	a*	b*	Hue Angle (°)	Chroma
Primary emulsions					
WPC	63.60 ± 2.68 ^b^	5.22 ± 0.56 ^a^	28.30 ± 0.63 ^a^	79.50 ± 1.28 ^c^	28.77 ± 0.55 ^a^
Ul-30	73.63 ± 1.91 ^a^	−1.51 ± 0.24 ^c^	19.24 ± 0.14 ^b^	94.47 ± 0.66 ^a^	19.33 ± 0.15 ^b^
En-30	70.93 ± 2.66 ^a^	1.99 ± 0.47 ^b^	16.30 ± 0.01 ^c^	83.00 ± 1.60 ^b^	16.43 ± 0.06 ^c^
Spray dried emulsions					
WPC	56.42 ± 0.75 ^a^	6.04 ± 0.46 ^a^	22.60 ± 2.33 ^a^	75.00 ± 1.45 ^b^	23.40 ± 2.23 ^a^
Ul-30	56.48 ± 1.01 ^a^	−0.25 ± 0.19 ^b^	14.10 ± 1.50 ^c^	91.00 ± 0.70 ^a^	14.10 ± 1.48 ^c^
En-30	56.55 ± 1.32 ^a^	0.63 ± 1.46 ^b^	18.22 ± 1.79 ^b^	88.28 ± 4.42 ^a^	18.27 ± 1.79 ^b^

Data are presented as mean ± standard deviation (*n* = 3), and values denoted by different letters within each column are significantly different (*p* < 0.05). WPC: Whey protein concentrate; Ul: Ultrasonicated; En: Hydrolyzed.

**Table 5 pharmaceutics-14-02434-t005:** Encapsulation efficiency (EE), size, and oxidative stability of spray-dried grape seed oil emulsions.

Treatment	EE (%)	Size (μm)			Peroxide Value
		Powder Particles	Primary Emulsions	Reconstituted Emulsions	
WPC	83.60 ± 2.52 ^b^	9.83 ± 0.84 ^a^	0.286 ± 9.61 ^a^	0.737 ± 10.15 ^a^	3.57 ± 0.55 ^a^
Ul-30	88.77 ± 1.62 ^a^	8.77 ± 0.47 ^b^	0.253 ± 9.07 ^b^	0.575 ± 23.46 ^b^	2.27 ± 0.21 ^b^
En-30	91.67 ± 2.63 ^a^	8.60 ± 0.79 ^b^	0.192 ± 10.02 ^c^	0.296 ± 17.01 ^c^	1.43 ± 0.45 ^b^

Data are presented as mean ± standard deviation (*n* = 3), and values denoted by different letters within each column are significantly different (*p* < 0.05). WPC: Whey protein concentrate; Ul: Ultrasonicated; En: Hydrolyzed.

## Data Availability

Not applicable.

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
