# Peer review of "Modification of Whey Proteins by Sonication and Hydrolysis for the Emulsification and Spray Drying Encapsulation of Grape Seed Oil"

_pharmaceutics, 2022, doi:10.3390/pharmaceutics14112434_

Round 1

Reviewer 1 Report

They prepared stabilized emulsion by whey protein which is commonly known as in milk protein. As the stabilization technique has been investigated in food region since a few decades, although it is not novel one, I think the characterization of emulsion may provide useful information. Actually, this manuscript should be submitted in Pharmaceutics journal because they did not use API, and should be submitted in food science region.

My complaint is overall lack of scientific explanation in discussion. They just describe the results alone, and there are few explanations which are satisfying and enlighten readers. This lack of explanation may lose the interest of readers.

In introduction, the authors mentioned the presence of peptide by enzymatic degradation, so they measured the hydrolysis of whey protein. But the contribution of peptide in the stabilization of emulsion was not clarified in this manuscript. They do not fulfill their responsibility of explanation.

There are inactivation processes of protein (heating incubation and spray drying at higher temperature). However, the effect is well-discussed.

Reviewer 2 Report

The manuscript carriers a v good idea of utilizing whey plant protein as an emulsifier and the use of a green method such as ultra-sonication in increasing the encapsulation of grape seed oil in the emulsion oil phase rather than the non-green chemical methods.

I recommend the publication of the manuscript after carrying the following:

1- Did the authors determine the type of the formed emulsion by any means such as the dying, conductivity or any other method?

2- The introduction should contain a paragraph about the advantages pf proteins as carriers of different molecules and compounds. The authors can refer to:

10.1021/acsomega.9b01580 AND 10.1016/j.ijpharm.2018.12.015

3- The nature of error bars in Figures 1,2, and 3 should be stated.

4- The error bars should be added to Figure 4(g).

Reviewer 3 Report

This manuscript entitled "Modification of whey proteins by sonication and hydrolysis for the emulsification and spray drying encapsulation of grape-seed oil" investigated the efficacy of whey protein concentrate (WPC) modified by ultrasonication or partial enzymolysis as emulsifiers, and evaluated spray drying encapsulation of grape-seed oil using WPC. The authors have performed well-designed experiments, and the analysis of the obtained data is appropriate to describe what they found. 

I have a few minor comments, explained below.

1. On page 6, line 11 from the bottom: ABTS inhibition rate (41.3%) of sonication for up to 45 min (Ul-45) is probably smaller than the value (>45%) shown in Fig.1a.

2. Fig. 1, Fig. 2, Fig. 3, and Fig. S1-S5: In these figures, Enzymolysis was abbreviated as "H", but I think it would be better to unify it with "En".

3. Fig. 4, Fig. 6, and Fig. 7: In these figures, the abbreviations "UWPC" and "HWPC" are used, but I think it would be better to use "Ul" and "En".

4. Table 2: Please set "a", "b", and "ab" in the PDI and zeta potential columns to superscript.

5. Figure 4: Please indicate which images (a-f, h)  in Figure 4 correspond to which samples (WPC, Ul-30, or En-30). I think that "plot (c)" in the title of Figure 4 is "plot (g)".

Round 2

Reviewer 1 Report

I do not have comments about the revised manuscript.